# An Experimental Study for Deriving Fire Risk Evaluation Factors for Cables in Utility Tunnels

**Hyun Jeong Seo \*** , **Yon Ha Chung and Tae Jung Song**

Department of Research Planning Division, KIT Valley, Seoul 08512, Republic of Korea;
yonhas@naver.com (Y.H.C.); tjsong73@gmail.com (T.J.S.)
* Correspondence: sincedeni@naver.com; Tel.: +82-2-2624-3550

**Abstract:** In this study, we performed three tests to measure the fire-retardant performance of power cables installed in utility tunnels. The standards we applied for testing are ISO 5660-1, NES 713, and IEEE 1202. Specifically, we performed a cone calorimetric analysis, calculated the toxicity index, and measured the flame spread length on material surfaces. Even though the same fire-retardant chemical composites were applied, various differences in fire-retardant performance were found depending on the chemical properties of the cable sheath and insulation. We also found that gaseous substances generated during the burning of cables can serve as important risk assessment factors in fires. We determined that, in addition to the heat generated when the cable burns, the toxic gases emitted at this time can also be a risk factor. This is because it is important to consider any potential risk to a person entering as part of an initial response to an event or accident involving cables installed in utility tunnels. Moreover, in the event of a fire in the cable, there is a risk of hazardous substances flowing into the city center as toxic gases are released. Therefore, we determined that the risk of hazardous gases emitted during cable fire should be reflected in the fire-retardant performance standard.

**Keywords:** cable fire; toxicity index; fire characteristics; utility tunnels; risk assessment

## 1. Introduction

A utility tunnel is a facility that includes various types of infrastructure such as power cables, communication cables, water and sewage pipes, and waste transport pipes. Utility tunnels are increasingly being used to optimally arrange supply piping, water supply, and sewage pipes, and other pipelines to improve city roads, traffic conditions, and urban planning [1–4]. However, there are certain dangerous situations that can arise after installing a utility tunnel underground [3,4]. For example, utility tunnels are particularly vulnerable to various disasters, and to fire in particular [4–6]. This is because utility tunnels house many combustible materials that can easily become ignition sources, such as cables, distribution panels, gas supply facilities, and communication facilities. Among these combustible materials, cables are most vulnerable to fire, and they can be a major ignition source [3,4]. Moreover, since the space is sealed in a utility tunnel, the extinguishing activities available in the event of a fire are very limited.

When a fire breaks out in a utility tunnel, a city's communication function, power supply function, gas supply function, etc., can be interrupted, thus resulting in paralysis of the city's main functions [3,6–8]. Most studies examining fires in utility tunnels have mainly focused on the temperature inside the space [7,8]. Additionally, partial fire characteristics were reflected by establishing a theoretical model that simplified the internal structure of the field and utility tunnels. Partial fire properties are related to the propagation of flames on the surface of a material. The flame spread length is the only such property that is currently reflected in the cable's fire-retardant performance standard. It is important to more comprehensively evaluate the fire risk of cables installed in utility tunnels, as overloading of cables represents the main cause of fires in utility tunnels. In addition to

disrupting a city's communication network and power supply, cable fires also generate toxic gases from the burning of the cable sheath, which interfere with firefighting activities [9,10]. These harmful gases are also released to the outside through ventilation, which may harm local residents.

Despite these risks, there are currently no specific evaluation criteria for the fire risk of cables installed in utility tunnels. The existing fire-retardant performance regulations of cables stipulates that they cannot be used below a certain performance level, which is ascertained by conducting a fire-retardant performance evaluation test [9–11]. However, when only the performance level is used as a criterion, only the flame is considered, so such an evaluation is limited by not considering all the risk factors of a cable fire. Issues such as the amount of heat released in the event of a cable fire and the type and amount of toxic gases are currently not included in regulations related to the fire-retardant performance of cables [11–13]. It is therefore necessary to update the risk factors that are included in these regulations to allow for the safer installation of cables in utility tunnels.

Analyzing previous studies shows that existing research related to cable fires applied to utility tunnels has mainly been focused on the methods that can be used to calculate the amount of heat generated from a flame or suppress the spread of a flame [2–10]. Bian et al. [2] designed a system using a sensor that detects initial heat in the event of a fire in cables. Ye et al. [3] designed fire accident scenarios in utility tunnels and simulated the fire spread speed. In this work, the important factors were considered to be the vertical and horizontal temperature, and the temperature flow of the smoke was also analyzed to identify the safe section in the tunnel. An et al. [6] tried to demonstrate the relationship between the combustion characteristics (HRR, THR, and MLR) and heat flux of an optical cable applied within the cable through a cone calorimeter analysis. Studies examining the toxicity of combustion products of cables in enclosed spaces have also been conducted [14–16]. Those studies analyzed the thermal characteristics and components of the hazardous gases of combustion products by utilizing power cables with specifications applied to utility tunnels. However, for power cables, which are the main source of ignition in the event of a fire in a tunnel, the standards regulating the fire-retardant performance have limitations in that the harmfulness of gas and the characteristics of thermal decomposition due to combustion are not reflected.

In this study, experiments were conducted to analyze the characteristics of cable fires, including the heat release rate, the types of toxic gases and emissions, and flame spread characteristics. Further, by analyzing the results, we summarized the risk factors of specific cable fires. The results of this study are expected to help conceptualize the fire risk evaluation factors of cables and be useful as basic research data.

## 2. Materials and Methods

### 2.1. Materials

The two cables used in this experiment represent those used for power and control in utility tunnels, power plants, and other facilities. Such cables are made up of a sheath, insulation, and core (electrical wires). The sheath of both cables in this experiment was made of polychloro rubber. The insulation components were different, as one was made using ethylene propylene rubber (EPR) material whereas the other was made using cross-linked polyethylene (XLPE). The core of both cables was made of copper. Table 1 lists the physical specifications of the cable specimens.

The test specimens were prepared by making three types of samples: First, the cable specimens were manufactured according to the ISO 5660-1 [11] and ISO 19702 [12] standards in sizes of 100 mm × 100 mm × 25 mm. Second, the cable specimens were fabricated by cutting three to five sheaths and accompanying insulation with a mass of 1 ± 0.025 g to meet the NES 713 [13] standard. Third, seven cable specimens with a cable diameter of 25 mm were prepared according to the IEEE 1202 [14] standard, and the length of each specimen was cut to 3.5 m.

**Table 1.** Physical specifications of the experimental specimens.

| Materials | | CR/EPR Cable | CR/XLPE Cable |
|---|---|---|---|
| Application | | Power and control | Control |
| Power capacity (kV) | | 0.6 | 0.6 |
| Thermal characteristics | | Thermosetting | Thermosetting |
| Cable diameter (mm) | | 25 | 25 |
| Fire-retardant treatment | | FR | FR |
| Characteristics of physical composition | Sheath | Polychloro rubber (CR) | Polychloro rubber (CR) |
| | Insulation | Ethylene propylene rubber (EPR) | Cross-linked polyethylene (XLPE) |
| | Core (Electric wires) | Copper | Copper |

*2.2. Methods*

Cable fire performance evaluation tests can generally be divided into four categories: heat resistance, flame retardancy, fire resistance, and toxicity determination. In Korea, the "IEEE Standard for Flame Testing of Cables for Use in Cable Tray in Industrial and Commercial Occupancies (IEEE Std 1202-1991)" is applied as a regulation related to proof of the fire retardancy of power and control cables. However, since IEEE Std 1202 (1991) only evaluates the spread of the flame, it has a limitation in that it does not reflect other combustion characteristics.

In the present study, three experiments were conducted to identify the fire characteristics of cables. First, cone calorimetric analysis was performed based on the ISO 5660-1 standard to identify the amount of heat released and the combustion characteristics in a fire. This experiment allows for the measurement and analysis of the heat release rate (HRR), total heat release (THR), time to ignition (TTI), and mass loss rate (MLR). To analyze the heat release from the cable covering material, insulation material, and copper wire as well as the emission characteristics of the combustion products, the temperature in the laboratory was set and maintained at $25 \pm 2$ °C with a relative humidity of $50 \pm 5\%$. The test was performed under conditions of a cone heater heat flux of 50 kW/m$^2$ and an exhaust flow rate of $0.024 \pm 0.002$ m$^3$/s. The test was conducted a total of three times for 20 min each time, and the results were summarized as the average value obtained. This experiment conformed to the ISO 5660-1 standard.

The second experiment was a quantitative toxic gas emission measurement experiment measuring the combustion products of the test specimens. The "Naval Engineering Standard (NES) 713 (1985)" is a technical standard that can be applied to the calculation of the combustion gas toxicity index of non-metallic materials such as small materials and cable components including the sheath and insulation materials. The size of the chamber in this experiment was 0.6 m$^3$, while the gas mixing ratio was set at 2 L/min for methane gas and 15 L/min for air. The flame height of the burner was maintained at approximately 100 mm, while the flow rate was adjusted so that the temperature of the flame was $1150 \pm 25$ °C. The test specimens exposed to the flame maintained a continuous flame contact state until reaching complete combustion. After combustion, the burner flame was ignited and the mixing fan was operated for 30 s to detect and sample the combustion gas in the chamber through each detector tube. After gas sampling, the residues inside the chamber were immediately discharged through a forced discharge device; this discharge lasted over three minutes.

Finally, an experiment was conducted to measure the flame spread properties of the test specimens. IEEE 1201 is a cable vertical tray test standard, and in this experiment, pre-treatment was performed for more than three hours at a temperature of 18 °C or higher in a constant temperature and humidity room before the test specimens were installed on the tray according to the corresponding standard. A vertical tray was installed

in a 2.4 m × 2.4 m × 2.4 m firebrick chamber, while the test specimens were fixed on a 30 cm × 2.4 m tray at 1/2 intervals of the cable diameter. The air flow rate was set to 1280 cm$^2$/s and the flame source was applied at 20 kW (approximately 70,000 BTU/h). The flame application time was set to 20 min in total, and the experiment was conducted.

Figure 1 shows the tray and equipment used for the cable flame test as specified by the IEEE. We provide explanations for specific sequences for the cable tray test below.

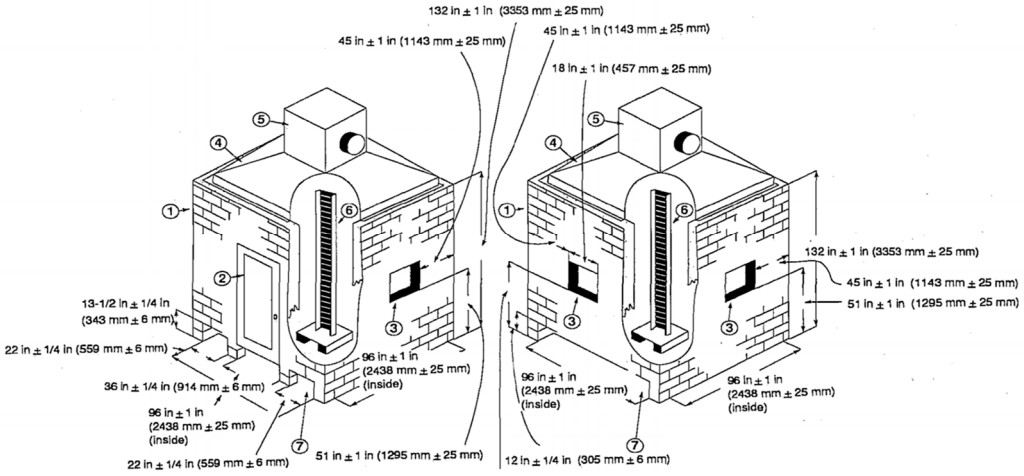

**Figure 1.** Flame Test Facility [14].

We provided a detailed explanation of the process by which the experiment is performed in Figure 1. First, this is an explanation of cables and cable trays. The cables are attached to the cable trays and pretreatment is performed. The tray is installed parallel to the wall of the chamber, and the cable is installed such that the end of the sample touches the top of the burner. Second, the flow rate of the exhaust duct is adjusted to be 0.65 m$^3$/s ± 0.02 m$^3$/s. Third, the burner is installed horizontally at an angle of 20 degrees and separated by 75 mm from the sample surface. This propane flow is theoretically equivalent to a 20 kW heat release, but the actual heat release is lower due to incomplete combustion of the burner. Fourth, the airflow rate is set to 1280 cm$^3$/s ± 80 cm$^3$/s under standard temperature and pressure. Finally, the flame of the burner continues to ignite the sample for 20 min. When the test is finished, the flame of the burner is turned off and the cable is left burning until it is self-extinguished.

## 3. Results

This section presents this study's findings regarding the combustion properties, toxicity gases emission characteristics, and flame spread length on the cable surface. First, the combustion properties obtained by analyzing the results of the cone calorimetry analysis experiment are presented. Following the ISO 5660-1 standard, this experiment was conducted three times with two test specimens, and the average value of these results is presented in this paper. Following the NES 713 standard, gaseous substances were collected through a total of two tests, after which the amount of gas emitted was measured and the toxicity index was calculated. The results of the experiment are presented as the average of two measurements and calculations. In this study, this was also applied and the average of the experimental results and calculated values was presented in this paper.

### 3.1. Combustion Properties

Two types of test specimens were used in this study. Specifically, cable E was classified as having an insulation material of EPR (ethylene propylene rubber) whereas cable X was classified as having an insulation material of XLPE (cross-linked polyethylene). Table 2 presents the cone calorimetry test results for these two types of cables.

**Table 2.** Results of cone calorimetric analysis.

| Parameter | Cable E | Cable X |
|---|---|---|
| PHRR (kW/m$^2$) | 149.06 | 124.65 |
| tPHRR (s) | 69 | 70 |
| THR (MJ/m$^2$) | 52.77 | 32.10 |
| Total oxygen consumed (g) | 31.13 | 18.73 |
| Time to ignition (s) | 38 | 36 |
| Initial mass (g) | 714.98 | 703.73 |
| Mass lost (g) | 58.77 | 38.48 |
| Remaining mass (g) | 656.21 | 665.25 |
| Mass loss rate (%) | 8.22 | 5.47 |

For the HRR, the peak of the HRR (PHRR) did not exceed 200 kW/m$^2$ for either of the test specimens. The times at which PHRR appeared (tPHRR) were 69 s for Cable E and 70 s for Cable X, which were almost identical. The THR value for Cable X was 32.10 MJ/m$^2$, which was 20 MJ/m$^2$ lower than that of Cable E. The ignition time (TTI) on the surface of the test specimens was 38 s for Cable E and 36 s for Cable X, thus showing no significant difference.

However, Cable E exhibited higher released heat values than Cable X. These results are attributed to the different insulation components of EPR and XLPE. Both cables were treated with fire-retardant additives on the sheath and insulation. No significant difference was observed in terms of the sheath, because the sheaths of both test specimens belonged to the CR series, but the degree of combustion was determined to be different due to the difference in insulation components [15,16]. Figure 2 shows the HRR and THR graphs obtained after the cone calorimetry analysis. From the graphs, it can be seen that two peaks appear in the HRR curve. This suggests a phenomenon wherein heat release is delayed due to the formation of a char layer after an increase in heat release due to the evaporation of volatile substances in the initial cable material. Then, as the surface of the char is broken, combustion proceeds again, and the heat release rate increases, which is the meaning of the second peak.

The results shown in the cable HRR curve confirm that the HRR of Cable E was not constant after the PHRR, as change occurred from 200 s until the end of the experiment. By contrast, Cable X's HRR curve showed no significant change after one PHRR. This heat release characteristic was also reflected in the THR curve. This is attributed to the EPR-based cable insulation.

EPR-based and XLPE-based insulation types are mainly used for manufacturing cables for utility tunnels or power plants because of their excellent heat and fire resistance. When making cables using these materials, vulcanizing agents such as sulfuric acid and bromine are added as fire-retardant chemical composites [9,10]. EPR has a low degree of deterioration, good flexural properties, and self-extinguishing properties because of the organic-group (R) attached to the Si-O framework. However, the mixing of a fire-retardant filler with a vulcanizing agent during cable manufacturing causes the affinity of the EPR insulation material to be lower than that of XLPE, so the non-uniformity of the material can serve as an obstacle to improving its fire-retardant performance [9,16].

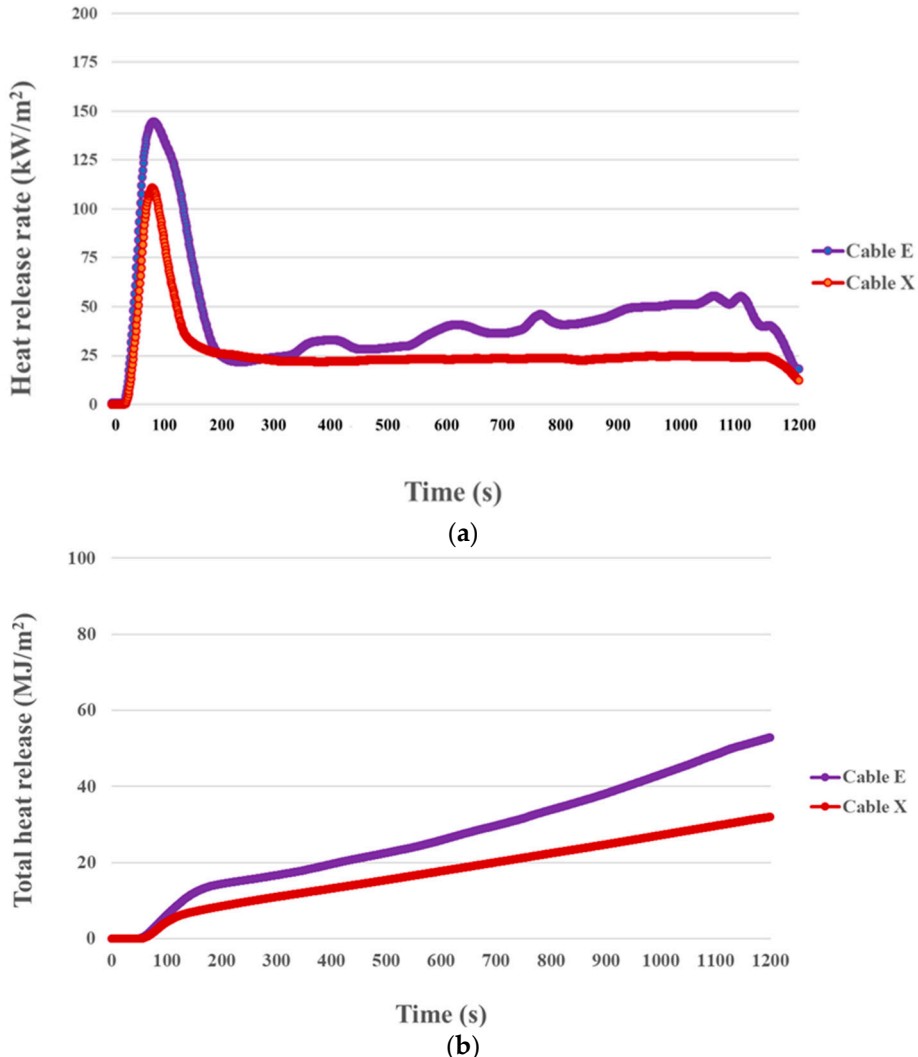

**Figure 2.** HRR and THR test results after the cone calorimeter test: (**a**) HRR results; (**b**) THR results.

Since the HRR represents the amount of heat released from the surface of a material, it may have limitations in terms of its ability to determine the degree of improvement in fire-retardant performance based on the size of the value. Therefore, we additionally compared the consumption of oxygen, which is a combustible gas that is used when the materials are burned. The oxygen consumption was 31.13 g, and the value consumed by Cable E was found to be higher than that of Cable X. When comparing the MLR values of the two specimens, the MLR value of Cable X was confirmed to be lower than that of Cable E. Although both types of cables were treated with fire-retardant additives, the fire-retardant performance differed depending on the components of the material itself.

### 3.2. Toxic Gas Emission Characteristics

To subdivide the risk by calculating the toxicity index (TI) of the gas generated during combustion, an experiment was conducted in accordance with the NES 713 standard. The test results were analyzed in accordance with the MIL-DTL standard [17,18]. The standard set by MIL-DTL is the TI, where the maximum value is 5.0 for the cable sheath and 1.5 for the cable insulation. In NES 713, the TI value is presented by applying the limit set by the MIL-DTL standard. Moreover, the critical values of thirteen gas components are set in detail. The specific types of gas are as follows: carbon dioxide ($CO_2$), carbon monoxide (CO), phenol ($C_6H_5OH$), ammonia ($NH_3$), hydrogen sulfide ($H_2S$), sulfur dioxide ($SO_2$), formaldehyde

(HCHO), hydrogen chloride (HCl), hydrogen bromide (HBr), hydrogen cyanide (HCN), nitrogen oxides ($NO_X$), hydrogen fluoride (HF), and acrylonitrile ($CH_2CHCN$).

Table 3 presents the combustion gases detected in Cable E and Cable X in this experiment.

**Table 3.** Combustion gas toxicity test results of experiment conducted in accordance with NES 713.

| Parameter | Cr (Critical Limit, ppm) | Cable E | | Cable X | |
|---|---|---|---|---|---|
| | | Sheath | Insulation | Sheath | Insulation |
| Carbon dioxide ($CO_2$) | 100,000 | 83,333 | 41,687 | 34,649 | 40,222 |
| Carbon monoxide (CO) | 4000 | 7321 | 6402 | 8373 | 6608 |
| Hydrogen sulfide ($H_2S$) | 750 | 0 | 0 | 0 | 0 |
| Ammonia ($NH_3$) | 750 | 0 | 0 | 0 | 0 |
| Formaldehyde (HCHO) | 500 | 30 | 119 | 29 | 115 |
| Hydrogen chloride (HCl) | 500 | 60 | 30 | 46 | 86 |
| Acrylonitrile ($CH_2CHCN$) | 400 | 12 | 36 | 23 | 40 |
| Sulfur dioxide ($SO_2$) | 400 | 655 | 30 | 144 | 57 |
| Nitrogen oxides ($NO_X$) | 250 | 149 | 71 | 87 | 230 |
| Phenol ($C_6H_5OH$) | 250 | 0 | 0 | 0 | 0 |
| Hydrogen cyanide (HCN) | 150 | 18 | 30 | 29 | 34 |
| Hydrogen bromide (HBr) | 150 | 20 | 10 | 15 | 29 |
| Hydrogen fluoride (HF) | 100 | 0 | 0 | 0 | 0 |

In total, nine toxic gases were emitted from Cable E and Cable X: dioxide, carbon monoxide, formaldehyde, hydrogen chloride, acrylonitrile, sulfur dioxide, nitrogen oxides, hydrogen cyanide, and hydrogen bromide. Comparing the combustion gases emitted from the cable sheath, the amount of carbon dioxide emitted from Cable E was higher than that emitted from Cable X. By contrast, the amount of carbon monoxide emitted from Cable X was higher than that emitted from Cable E. Carbon monoxide is produced during incomplete combustion, and its production indicates higher fire-retardant performance. Regarding the emissions of sulfur dioxide and nitrogen oxides, higher values were measured in Cable E. This was attributed the difference in the component ratios of the fire-retardant chemical composites processed on the cable sheath.

Among the released toxic gases, HCl can cause eye or upper airway mucosal irritation and airway damage, which can lead to suffocation [9,19,20]. As thermal decomposition progresses during the combustion of materials that contain components such as polyethylene and polyvinyl chloride, HCL is considered to be a product that is generated by organic acids and aromatic substances, among the chemical species included in the cable sheath [21,22].

Both cable sheaths are composed of polychloroprene, but there was confirmed to be a difference in the combustion products. This was attributed to the types of additives that are additionally used when polychloroprene is applied to the cable sheath. Polychloroprene is mainly used as a cable sheath because of its excellent insulation and fire-retardant properties. However, since it hardens at low temperatures, certain amounts of additives, typically sulfuric acid or bromine, are used to improve the durability of materials. When these additives are applied, a synergy effect is expressed in the fire-retardant effect [23]. However, when these components are burned, they can cause the emission of toxic gases such as hydrogen sulfide, sulfur dioxide, and hydrogen bromide. Therefore, it was determined that the additives used in the manufacture of cable sheaths require additional hazard consideration.

The combustion gas of the cable insulation was found to be similar to that of the cable sheath in terms of hazardous components. The amount of carbon monoxide emitted among the hazardous components exceeded the critical limit. Moreover, the emission of

formaldehyde from the cable insulation was found to be higher than that from the cable sheath. Notably, a large amount of nitrogen oxide was detected in the insulation of Cable X. Since the critical limit value stipulated in NES 713 is 250 ppm, the detection amount of nitrogen oxides in this experiment is not a problem in terms of regulations. However, in terms of the risk of nitrogen oxides, we determined that the 230 ppm detected in this experiment was dangerous because it was very close to the critical limit. Table 4 presents the combustion gas toxicity index calculation results.

**Table 4.** Toxicity index of the cable specimens.

| Parameter | Cable E | | Cable X | |
|---|---|---|---|---|
| | Sheath | Insulation | Sheath | Insulation |
| Carbon dioxide ($CO_2$) | 0.83 | 0.42 | 0.35 | 0.40 |
| Carbon monoxide (CO) | 1.83 | 1.60 | 2.09 | 1.65 |
| Hydrogen sulfide ($H_2S$) | 0.00 | 0.00 | 0.00 | 0.00 |
| Ammonia ($NH_3$) | 0.00 | 0.00 | 0.00 | 0.00 |
| Formaldehyde (HCHO) | 0.06 | 0.24 | 0.06 | 0.23 |
| Hydrogen chloride (HCl) | 0.12 | 0.06 | 0.09 | 0.17 |
| Acrylonitrile ($CH_2CHCN$) | 0.03 | 0.09 | 0.06 | 0.10 |
| Sulfur dioxide ($SO_2$) | 1.64 | 0.08 | 0.36 | 0.14 |
| Nitrogen oxides ($NO_X$) | 0.60 | 0.28 | 0.35 | 0.92 |
| Phenol ($C_6H_5OH$) | 0.00 | 0.00 | 0.00 | 0.00 |
| Hydrogen cyanide (HCN) | 0.12 | 0.20 | 0.19 | 0.23 |
| Hydrogen bromide (HBr) | 0.13 | 0.07 | 0.10 | 0.19 |
| Hydrogen fluoride (HF) | 0.00 | 0.00 | 0.00 | 0.00 |

*3.3. Flame Spread Characteristics*

The carbonized lengths of Cable E and Cable X were measured. In detail, the cable specimens were exposed to flame for a total of five times, for 20 min each time. Upon completion of the flame exposure test, the carbonization length was measured. The average carbonization length of Cable E was approximately twice that of Cable X. Based on this result, it was determined that the fire-retardant performance of Cable X was better than that of Cable E in terms of flame spread among fire-retardant performances. These results are attributed to the component characteristics of the cable specimens. Among the components of cable insulations used as fire-resistant fillers, XLPE does not lose the insulation properties of cables and can exhibit resistance to extreme fire. Although the EPR of Cable E also has fire resistance, its combustion progressed relatively more than that of Cable X because XLPE has stronger resistance to thermal decomposition due to cross-linking. Table 5 shows the results of flame spread tests.

**Table 5.** Results of flame spread experiments.

| Category | | Carbonization Length (mm) | | |
|---|---|---|---|---|
| | | Test 1 | Test 2 | Test 3 |
| Cable E | Cable E-1 | 900 | 910 | 935 |
| | Cable E-2 | 1010 | 910 | 1010 |
| | Cable E-3 | 1020 | 910 | 1010 |
| | Cable E-4 | 1000 | 910 | 1005 |
| | Cable E-5 | 970 | 760 | 990 |

**Table 5.** *Cont.*

| Category | | Carbonization Length (mm) | | |
|---|---|---|---|---|
| | | Test 1 | Test 2 | Test 3 |
| Cable X | Cable X-1 | 580 | 560 | 485 |
| | Cable X-2 | 570 | 560 | 500 |
| | Cable X-3 | 575 | 605 | 530 |
| | Cable X-4 | 595 | 650 | 550 |
| | Cable X-5 | 560 | 670 | 560 |

## 4. Conclusions and Implications

In this study, we measured and analyzed the fire characteristics of power cables used in utility tunnels and the toxicity of combustion gases. To measure the fire characteristics of cable specimens, cone calorimeter analysis was performed, and to measure the toxicity index of combustion products, the NES 713 standard was applied for the absorption of combustion gases using the Drager tube. We also measured the carbonization length to confirm the characteristics of flame spread on the surface of cable test specimens. Based on an analysis of the experimental results obtained in this study, we have derived the following implications:

First, when determining the fire-retardant level of power cables applied to utility tunnels, it is important for fire characteristics of cables to be specifically reflected. This is because the fire-retardant performance of a cable varies depending on the chemical composites of the sheath and insulation that make up the cable. According to the components of these chemical composites, we confirmed through experiments that fire resistance can differ even when using the same fire-retardant additives for treatment. It is also necessary to check the temperature at which the cable burns and the temperature at which it deteriorates, rather than simply considering the amount of heat emitted.

We determined that there is a limit to determining improvements in fire-retardant grade by simply considering reductions in the flame spread on the material surface or reductions in the carbonization length in the case of cable fire. We thought it should be considering maintaining or losing the function of power cables. The current fire-retardant performance standard of cables has a limitation in that it does not consider situations in which various functions, such as the power supply of cables, maintenance of facility functions, and maintenance of networks with the outside, might be lost. Considering these factors, the secondary risk due to functional loss is as dangerous as the temperature of the fire. Therefore, it is necessary to subdivide the grade by considering the chemical composites of the cable, the system, and the functions.

Second, the toxicity of combustion products should be considered when distinguishing fire-retardant levels in cable fires. When a fire occurs in utility tunnels, hazardous gases are generated, and workers inspecting the inside of the tunnel may be exposed to these harmful gases. These risks can lead to emergencies. Moreover, the release of a large amount of harmful gas can cause problems that affect firefighter safety during firefighting. We also note that utility tunnels are installed in downtown areas, and that it is important to consider that dangerous situations may occur if the air inside the utility tunnel leaks to the outside through the ventilation, thus harming people in the city.

Although we derived the above implications, our study has some limitations. First, while burning the cables, we measured the amounts of toxic gases emitted for 30 min, but time-series data on how the combustion gases were emitted in real time could not be confirmed. We determined that time-series data related to combustion gas emission are an important concept because plans for evacuation from utility tunnels or fire suppression plans can be set up differently depending on whether a lot of gaseous substances are released at the beginning of combustion or after some period of time. Further, since the toxicity index is a calculated result for an enclosed space, it is necessary to conduct

experiments that consider various space conditions. Since certain sections of utility tunnels are straight sections and involve mechanical ventilation, the spread of smoke or flame is an important risk assessment factor. Moreover, due to the spatial characteristics of utility tunnels, in the case of spaces such as T-shaped vents or natural ventilation vents, there are differences in airflow, so we determined that simulations reflecting spatial conditions would be necessary.

While carrying out this study, we were able to secure data on the derived fire characteristics of power cables in utility tunnels and the emission characteristics of hazardous gases. In future studies, we plan to perform modeling and fire simulations targeting utility tunnels based on the data obtained in this study. The data acquired in this study are expected to help elucidate the characteristics of the flow and diffusion of heat and smoke according to the internal shape of the utility tunnels.

**Author Contributions:** Conceptualization, H.J.S. and Y.H.C.; methodology, H.J.S.; software, H.J.S. and Y.H.C.; validation, H.J.S., Y.H.C. and T.J.S.; formal analysis, H.J.S.; investigation, Y.H.C.; resources, Y.H.C. and T.J.S.; data curation, Y.H.C. and T.J.S.; writing—original draft preparation, H.J.S., writing—review and editing, Y.H.C. and T.J.S.; visualization, Y.H.C.; supervision, H.J.S.; project administration, T.J.S.; funding acquisition, T.J.S. All authors have read and agreed to the published version of the manuscript.

**Funding:** This research was supported by the Institute of Information & Communications Technology Planning & Evaluation (IITP), with a grant funded by the Korea government (MSIT, MOIS, MOLIT, and MOTIE) (No. 2020-0-00061, Development of integrated platform technology for fire and disaster management in underground utility tunnel based on digital twin).

**Institutional Review Board Statement:** Not applicable.

**Informed Consent Statement:** Not applicable.

**Data Availability Statement:** Not applicable.

**Conflicts of Interest:** The authors declare no conflict of interest.

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
