# Peer review of "An Experimental Study for Deriving Fire Risk Evaluation Factors for Cables in Utility Tunnels"

_fire, doi:10.3390/fire6090342_

Round 1
Reviewer 1 Report
The paper is well structured. It addresses an important structural problem, the study for deriving fire risk evaluation factors for cables in utility tunnels. However, some major revisions are needed:
The abstract should include a statement of the problem you are trying to solve and the purpose of your research.
In the introduction, it is clear that the authors do not have enough information on the existing studies
The novelty of the paper should be described in the introduction.
The quality of the experimental results is good, but the Authors should add new findings from their study and differences from others' results in the Discussion section.
Conclusion has to be improved.

Author Response
I sincerely thank you for your review comments.
I wrote the response in the attached file, so please check it.

Reviewer 2 Report
Overall the authors have done a good job at explaining the experimental procedure and the results obtained. Following are some points that require some attention
1. There are grammatical mistakes through the paper and in some places the language used is difficult to follow. For example, line 209, "Since the critical limit value is 250 ppm, it is not a value beyond the standard value, but it is determined to have a risk because 230 ppm of the gas is emitted". What is the meaning of this sentence? A through proof reading should help improve these mistakes.
2. Was the data reported in this article collected from a single experiment? If multiple experiments were conducted, why not report the standard deviation and the mean in measured values.
3. It is confusing to understand the setup used in experiment 3, i.e., the experiment to measure flame spread properties. Consider including a pictorial representation of the experimental setup.
There are several grammatical mistakes throughout the paper. A through proof reading is recommended.
Author Response

(The authors gave the same response as above.)

Reviewer 3 Report
Really good work, but there are some lacks on the scientific side. The tests are correctly reported and the results too, but the reason of the cable behavior and of the tests are not enough reported.
Author Response

(The authors gave the same response as above.)

Round 2
Reviewer 1 Report
The manuscript is acceptable for publication